# Cell-Adapted Mutations and Antigenic Diversity of Influenza B Viruses in Missouri, 2019–2020 Season

**DOI:** 10.3390/v13101896

**Published:** 2021-09-22

**Authors:** Cynthia Y. Tang, Karen Segovia, Jane A. McElroy, Tao Li, Minhui Guan, Xiaojian Zhang, Shamita Misra, Jun Hang, Xiu-Feng Wan

**Affiliations:** 1Center for Influenza and Emerging Infectious Diseases, University of Missouri, Columbia, MO 65211, USA; ct3dg@health.missouri.edu (C.Y.T.); karens_122@outlook.com (K.S.); mg1484@msstate.edu (M.G.); xiaojian.zhang@missouri.edu (X.Z.); 2Department of Molecular Microbiology and Immunology, School of Medicine, University of Missouri, Columbia, MO 65211, USA; 3Bond Life Sciences Center, University of Missouri, Columbia, MO 65211, USA; 4Institute for Data Science and Informatics, University of Missouri, Columbia, MO 65211, USA; 5Department of Family and Community Medicine, School of Medicine, University of Missouri, Columbia, MO 65211, USA; mcelroyja@health.missouri.edu (J.A.M.); misras@health.missouri.edu (S.M.); 6Viral Diseases Branch, Walter Reed Army Institute of Research, Silver Spring, MD 20910, USA; tao.li.ctr@mail.mil (T.L.); jun.hang.civ@mail.mil (J.H.); 7Department of Electrical Engineering & Computer Science, College of Engineering, University of Missouri, Columbia, MO 65211, USA

**Keywords:** influenza, respiratory diseases, influenza vaccines, influenza B virus, influenza viruses type B, antigenic drift, reassortment

## Abstract

Influenza B viruses (IBVs) are causing an increasing burden of morbidity and mortality, yet the prevalence of culture-adapted mutations in human seasonal IBVs are unclear. We collected 368 clinical samples from patients with influenza-like illness in Missouri during the 2019–2020 influenza season and recovered 146 influenza isolates including 38 IBV isolates. Of MDCK-CCL34, MDCK-Siat1, and humanized MDCK (hCK), hCK showed the highest virus recovery efficiency. All Missourian IBVs belonged to the Victoria V1A.3 lineage, all of which contained a three-amino acid deletion on the HA protein and were antigenically distant from the Victoria lineage IBV vaccine strain used during that season. By comparing genomic sequences of these IBVs in 31 paired samples, eight cell-adapted nonsynonymous mutations were identified, with the majority in the RNA polymerase. Analyses of IBV clinical sample–isolate pairs from public databases further showed that cell- and egg-adapted mutations occurred more widely in viral proteins, including the receptor and antibody binding sites on HA. Our study suggests that hCK is an effective platform for IBV isolation and that culture-adapted mutations may occur during IBV isolation. As culture-adapted mutations may affect subsequent virus studies and vaccine development, the knowledge from this study may help optimize strategies for influenza surveillance, vaccine strain selection, and vaccine development.

## 1. Introduction

The seasonal influenza virus is responsible for 290,000–650,000 human deaths worldwide each year [1]. These annual epidemics are caused by the co-circulation of influenza A viruses (IAVs) and influenza B viruses (IBVs). The influenza A subtypes causing contemporary seasonal outbreaks are primarily composed of subtypes H1N1 and H3N2 viruses, which have been co-circulating in humans since 1977. The H1N1 subtype caused the 1918 pandemic and circulated in humans until 1957 [2], then re-emerged and caused seasonal outbreaks from 1977 through 2009 [3]. This was later replaced with a novel swine-origin H1N1 virus, resulting in the 2009 pandemic [4,5]. The H3N2 subtype began circulating in 1968 and caused the 1968 pandemic [6,7]. Seasonal influenza B, identified in 1940 [8], comprises two lineages, B/Yamagata/16/88-like [Yamagata] and B/Victoria/2/87-like [Victoria], which have been co-circulating since at least 1983 [9,10]. While IAVs have been responsible for the majority of seasonal influenza complications, IBVs are causing considerably high burdens of morbidity and mortality [11,12,13,14].

Both genetic and antigenic drift facilitate the annual recurrence of influenza epidemics [15]. As an RNA virus, both IAVs and IBVs have a high mutation rate, ranging from 0.4 × 10^−3^ to 2.0 × 10^−6^ mutations per nucleotide per year, leading to genetic drift, because the RNA-dependent RNA polymerase (RdRp) lacks the 3′ to 5′ exonuclease proofreading capability. Antigenic drift is caused by evolutionary pressures that lead to an accumulation of mutations, particularly at the hemagglutinin (HA) and neuraminidase (NA) genes [16]. IBVs evolve more slowly compared with IAVs [17,18,19]. Of the IBV lineages, the Victoria lineage is undergoing more rapid evolution with greater positive selection pressure than the Yamagata lineage [17]. Structurally, the Yamagata virus binds to the host alpha-2,6-linked sialic acid receptors, whereas the Victoria virus binds to both alpha-2,3 and alpha-2,6 receptors, contributing to their differences in antigenic drift events [20,21].

Annual influenza vaccinations serve as a primary option for the prevention and control of influenza viruses [22,23]. Current vaccines are composed of two influenza A strains (H1N1 and H3N2) and either one (trivalent) or two (quadrivalent) influenza B strains [24]. Antigenic drift can cause low efficacy of seasonal influenza vaccines and necessitate annual vaccine reformulations. Due to antigenic drift, seasonal vaccine strains have been updated almost every year, especially during the past 10 years. Comparatively, IAVs have a much higher frequency of antigenic drift events than IBVs, as is reflected with the less frequent updates of the IBV component in the seasonal influenza vaccine.

Conventional serological tests such as hemagglutinin inhibition assays used in antigenic analyses for influenza vaccine strain selection require large virus quantities, and thus, the viruses often need to be isolated and propagated in Madin–Darby canine kidney (MDCK) cell lines or embryonated chicken eggs [25]. This can lead to cell- or embryonated egg-adapted mutations during viral propagation, particularly at the HA and NA surface glycoproteins [26,27,28,29,30]. Some prior studies have suggested that viruses propagated on MDCK cells have resulted in better antigenic matches than those grown on eggs [31]. During the seasons of 2012–2013 (VE = 49%, 43,000 deaths), 2016–2017 (VE = 40%, 38,000 deaths), and 2017–2018 (VE = 38%, 61,000 deaths), egg-adaptive mutations in the HA protein caused challenges in developing antigenically matched vaccine strain with circulating strains [32,33,34,35,36]. However, most of these studies on culture-adapted mutations were on IAVs, and the frequencies and implications of cell- or egg-adaptations in IBVs are yet to be fully understood.

In this study, we collected clinical samples from Columbia, Missouri during the 2019–2020 influenza season, particularly in the influenza epidemic months of February and March of 2020, to study the genetic and antigenic evolution of IBVs in humans and to assess whether passage of clinical samples in cells induces adaptive mutations across viral genes. Specifically, we performed serological analyses to characterize antigenic properties of the circulating influenza virus strains during this period and genomic analyses to determine the genetic changes that occurred during the influenza outbreaks during the 2019–2020 season in Missouri and whether culture-adapted mutations occurred during virus isolation.

## 2. Materials and Methods

Clinical samples: Nasopharyngeal swab specimens (*N* = 368) were collected from patients with influenza-like illness, who visited any of three urgent care facilities in Columbia, Missouri between February and March 2020. Samples were collected using a BD™ Universal Viral Transport Kit (3 mL, BD, Franklin Lakes, NJ, USA) and stored at −70 °C.

Nucleic acid extraction: RNA extraction was performed using the MagMAX^TM^ Pathogen RNA/DNA kit (Thermo Fisher Scientific Baltics, Vilnius, Lithuania) following the manufacturer’s instructions. The nucleic acid was resuspended in 90 μL of elution solution.

Quantitative reverse transcription polymerase chain reaction (qRT-PCR): The commercial kit AgPath-ID^TM^ One-Step RT-PCR (Ambion, Austin, TX, USA) and CDC influenza A/B primers were used for screening the samples for influenza viruses in a single reaction in a QuantStudio^TM^ 6 Flex Real time PCR System (Life Technologies, Paisley, UK). A 25 μL reaction containing 12.5 μL of the 2X reaction buffer, 0.5 μL of each primer (50 μM stock), or probe (10 μM stock) was used to reach a final concentration of 1 μL of the 25X RT-PCR enzyme mix. Additionally, 1.67 μL of detection enhancer were used.

Cell lines: Humanized MDCK cells (hCK) were kindly provided by Dr. Yoshihiro Kawaoka from the University of Wisconsin-Madison, USA. The MDCK-Siat1 cells were kindly provided by Dr. Mikhail Matrosovich from Philipps University, Germany. The MDCK-CCL34 cells were obtained from the Biodefense and Emerging Infections Research Resources Repository. All cells were maintained until use at 37 °C under 5% CO_2_ in Dulbecco’s modified Eagle medium (Gibco DMEM; Thermo Fisher Scientific, Waltham, MA, USA) supplemented with 10% fetal bovine serum (Gibco; Thermo Fisher Scientific Baltics, Vilnius, Lithuania). These three cell lines were selected, because hCK has overexpression of α-2,6-sialoglycans and almost completely deleted expression of α-2,3-sialoglycans, and hCK has been demonstrated to be a more efficient cell line for influenza B, A/H1N1pdm, and even more efficient for A/H3N2 virus isolation and propagation than other MDCK cells for influenza viruses [37]. MDCK-Siat1 has overexpression of α-2,6-sialyltransferase but a marginal decrease in α-2,3-sialylation [38]. Similar to hCK cells, Siat-1 has been shown to improve isolation rates over conventional MDCK cell lines (CCL34) in influenza B, A/H1N1, and especially in A/H3N2 viruses [39]. Finally, MDCK-CCL34 cells express higher levels of α-2,3-syaloglycans compared with α-2,6-syaloglycans and are widely used for influenza virus isolation and propagation, and they exhibit relatively low levels of both α-2,3- and α-2,6-sialic acids with slightly higher levels of α-2,3-sialylation compared with α-2,6-sialylation [40].

Virus isolation: Clinical samples were diluted 1:10 in Opti-MEM media containing a 1X final dilution of antibiotic–antimycotic solution for 30 min at 4 °C. A total volume of 0.5 mL was inoculated in T25 flasks containing confluent cells in Opti-MEM media with antibiotics–antimycotics at 35 °C for 3 to 4 days until 75% cytopathic effect (CPE) was observed. Supernatant was collected and centrifuged at 4000 revolutions per minute, and the hemagglutination capacity was tested using 0.5% solution of turkey erythrocytes.

Ferret antisera: The following sera were obtained from the Influenza Research Reagent Resources: FR-1617 B/Colorado/6/2017, FR-1081 B/Nevada/03/2011, FR-1305 B/Texas/02/2013, FR-392 B/Brisbane/60/2008, B/Washington/02/2019 (egg), FR-810 B/Wisconsin/1/2010.

Hemagglutination inhibition (HAI) assays: Isolated viruses were characterized antigenically following previously described protocols [41].

Antigenic cartography: HAI results were visualized by AntigenMap (https://sysbio.missouri.edu/software/AntigenMap/, accessed on 19 July 2021) [42,43,44]. A titer of 20 was used as the low reactor threshold and a low rank (rank = 2) matrix completion was used to minimize noise from the HAI data. A single unit on the antigenic map represents a log_2_ unit in HAI titers.

Next generation sequencing: Whole genome RT-PCR amplification was conducted using 12 pairs of primers [45,46,47]. Amplicon libraries were prepared using the Nextera DNA Flex Library Prep kit, then sequenced with MiSeq Reagent Kit v3 (600 cycles) and MiSeq sequencing system (Illumina, San Diego, CA, USA). The resulting paired-end reads were analyzed for quality, mapped to a reference genome (B/Utah/17/2020, accession: MT499571-MT499578), and assembled using CLC Genomics Workbench v21.0.3 (Qiagen, Hilden, Germany). Pairwise analyses were conducted using CutAdapt v3.4 (National Bioinformatics Infrastructure Sweden, Uppsala, Sweden) [48], Bowtie2 v2.4.4 (Johns Hopkins University, Baltimore, Maryland, USA) [49], and samtools v1.13 (https://github.com/samtools/, accessed on 4 May 2021), and polymorphisms were identified using pysamstats v1.1.2 (https://github.com/alimanfoo/pysamstats, accessed on 4 May 2021) and confirmed using the Integrated Genomics Viewer (https://software.broadinstitute.org/software/, accessed on 10 June 2021). Cell-adapted mutations were verified manually using BioEdit v7.2.5 [50]. Total reads ranged from 142,724 to 2,161,170 for each sequence, with an average of 1,092,806 reads (standard deviation (SD) = 751,073). The average number of reads for clinical samples was 1,725,216 (SD = 239,826), and the average number of reads for the isolates was 460,396 (SD = 520,405). GenBank accession numbers for these 31 IBV isolates are MZ951177–MZ951671.

Phylogenetic analyses: The basic local alignment search tool (BLAST) was used to identify sequences with the closest identities to each sample, and the top 20 sequences across all segments were included in each phylogeny. Reference genomes from the Victoria lineage (B/Rhode Island/01/2019, EPI_ISL_409247; B/Nigeria/3352/2018, EPI_ISL_330532; B/Missouri/12/2018, EPI_ISL_309120; B/Hong Kong/286/2017, EPI_ISL_276541; B/Florida/78/2015, EPI_ISL_207237; B/Brisbane/60/2008, EPI_ISL_246494; B/Brisbane/46/2015, EPI_ISL_219054; B/Hong Kong/269/2017, EPI_ISL_276540; B/Florida/103/2016, EPI_ISL_248973; B/New Jersey/1/2021, EPI_ISL_246881; B/Colorado/6/2017, EPI_ISL_277231; B/Nevada/03/2011, EPI_ISL_94747; B/Florida/78/2015, EPI_ISL_207237; B/Texas/02/2013, EPI_ISL_139913; B/Washington/02/2019, EPI_ISL_341131) and the Yamagata lineage (B/Phuket/3073/2013, EPI_ISL_517766; B/Texas/81/2016, EPI_ISL_237464; B/New Hampshire/01/2018, EPI_ISL_296628) were also included in the phylogenetic trees for each segment. These sequences were downloaded from the Global Initiative on Sharing Avian Influenza Data (GISAID) consortium EpiFlu database (http://www.gisaid.org, accessed on 2 July 2021).

All selected sequences were aligned using MUltiple Sequence Comparison by Log-Expectation (MUSCLE8, European Bioinformatics Institute, Cambridgeshire, United Kingdom). Phylogenetic analyses were performed with BEAST2 (https://www.beast2.org/, accessed on 22 July 2020) with a Hasegawa–Kishino–Yano (HKY) substitution empirical model, strict clock model, and coalescent constant population prior. Tracer v1.7.1 was used to evaluate phylogenetic quality using a cutoff of 200 for the effective sample size (ESS) to assess convergence. The consensus tree was generated using TreeAnnotator v2.6.3.0 and visualized with FigTree v1.4.4 (http://tree.bio.ed.ac.uk/software/figtree/, accessed on 2 February 2021).

## 3. Results

### 3.1. Virus Isolation in MDCK-hCK Cell Lines Showed the Highest Virus Recovery Rate

Of the 368 samples, 146 (39.7%) tested positive for influenza with 38 samples (10.3%) positive for IBV and 108 (29.3%) for IAV by qRT-PCR. Thirty-one of the IBV samples were positive to virus isolation in hCK cells. Other MDCK cell lines such as CCL34 and SIAT1 were also used to compare cell sensitivity to virus isolation. The MDCK-hCK cell lines showed the highest performance for both IAV and IBV following the first passage of the viruses with an isolation rate of 87.96% and 81.58%, respectively, compared with the CCL34 (42.12% and 57.89%, respectively) and SIAT1 (71.30% and 57.89%, respectively) cell lines (Table 1).

### 3.2. Culture-Adapted Mutations Were Identified in IBVs

To identify culture-adapted mutations, we sequenced 38 IBVs from the IBV-positive clinical swab samples. Of the 38 paired swab and isolate (hCK derived; first passage) samples subjected to MiSeq sequencing, 31 paired samples yielded complete genomic data and were included in the analysis to determine hCK cell-adapted mutations and polymorphisms. The genomes for the other seven pairs were either incomplete or with a low coverage and thus not included our subsequent analyses. From these 31 paired samples, we identified 13 cell-adapted mutations in total among nine viruses, ranging from one to three mutations in each virus, and each of these mutations was unique to each virus. We detected eight nonsynonymous mutations among these paired samples that have not been previously published (HA-R581C; PB2-T46S, PB2-V98G; PB1-E112G, PB1-R350G, PB1-V501A, PB1-L674Q; and NP-E140A), six of which were from the RNA polymerase complex (Figure 1) and five synonymous mutations (Table 2). Polymorphisms were also identified in PB2-T293G, PB2-T2097C, PB1-A255G, PB1-A1048G, PB1-T1502C, and NP-A419C (Table 2).

In addition to our samples, we analyzed all Victoria lineage IBV samples available on GISAID with a sequence from an original specimen and at least one cell or egg sequence to determine whether culture-adapted mutations also occurred among samples submitted to a public database. Passage information is not consistently reported in the GISAID strains and, thus, was not included in the analysis. Of the initial 91,884 distinct and complete Victoria lineage sequences on GISAID as of 2 July 2021, 539 strains contained at least one clinical sample–isolate pair. Pairwise comparison of these strains revealed multiple additional cell- and egg-adapted mutations (Table 3). Three distinct mutations were identified at PB2, one at PB1, two at PA, two at NS1, two at NP, two at NEP, one at NB, 11 at NA, and eight at HA. Notably, on the HA protein, 14 strains contained a cell- or egg-adapted mutation at position 212 and six strains contained a mutation at position 214 (Figure 2). Additionally, of the complete Victoria IBV sequences with paired isolates, 10.3% contained culture-adapted mutations (6.7% cell, *N* = 537; 38.3% egg, *N* = 60).

### 3.3. Antigenic and Genetic Diversity of Missourian IBV in the 2019–2020 Influenza Season

Phylogenetic analyses showed that the IBVs circulating during the 2019–2020 influenza season belonged to the Victoria lineage, all in the clade V1A.3-3DEL. Additionally, all samples contained three amino acid deletions (3DEL) at positions 162–164 of the HA protein sequences when compared with B/Brisbane/60/2008 (EPI_ISL_246494) as shown in Figure 3. Of note, the IBV vaccine strain used in the 2019–2020 influenza season (B/Colorado/06/2017) had two amino acid deletions (2DEL).

### 3.4. Antigenic Diversity of Missourian IBVs in the 2019–2020 Influenza Season

To test the antigenic diversity of the IBVs, hemagglutination inhibition assays and antigenic cartography were performed for 32 Missouri isolates and 11 reference strains against seven ferret sera for six representative IBVs (B/Colorado/6/2017, B/Nevada/03/2011, B/Texas/02/2013, B/Brisbane/60/2008, B/Washington/02/2019, and B/Wisconsin/1/2010). The sera for egg and cell isolates of B/Washington/02/2019 were included (Table 4). Antigenic analyses suggested all Missourian isolates reacted well to the Victoria lineage IBV-derived ferret sera but poorly with the Yamagata lineage IBV-derived ferret sera. Of all ferret sera, all Missourian isolates had the highest HI titers against B/Washington/02/2019, which also has the 3DEL, but 4- to 8-fold lower titers against B/Colorado/06/2017, which is the vaccine strain used during the 2019–2020 season and contains the 2DEL [51]. Of note, these Missouri isolates reacted better to the ferret antisera against the cell isolate than the egg isolate of B/Washington/02/2019.

Antigenic cartography showed that these Missouri isolates have an average antigenic distance of 1.60 ± 0.22 units from B/Colorado/06/2017 (Figure 4), suggesting these isolates are antigenically distinct from the vaccine used in this season. On the other hand, the average antigenic distances within Missourian isolates were 0.40 ± 0.21 units (Figure 4). Each unit represents a 2-fold change in hemagglutination inhibition titers.

## 4. Discussion

Influenza B viruses are endemic around the world, leading the World Health Organization (WHO) to recommend the inclusion of Yamagata and Victoria strains in the seasonal influenza vaccine [24]. While the use of clinical swab samples for vaccine development would be ideal in minimizing cell- and egg-adapted mutations, this is not yet feasible. Previous studies have identified MDCK cells to promote the most viral growth when compared with Vero cells, both of which have been used frequently in influenza studies [25]. We extended this finding by testing our sample growth on various MDCK cell lines, including hCK, CCL34, and SIAT1 and found that the hCK cell lines showed the highest performance for both IAV and IBV following the first passage, consistent with findings by Takada et al. [37]. Higher replication efficiency results in fewer required passages and likely fewer cell adaptations. This finding suggests hCK cell lines are an effective platform for virus isolation and propagation and potentially allow for fewer culture adaptations during vaccine development.

During the 2019–2020 influenza season in the US, influenza caused 400,000 hospitalizations and 22,000 deaths [52]. Notably, influenza B and predominantly the B/Victoria lineage (>99% of IBVs) emerged unusually early in the season and made up approximately half of all influenza cases compared with 10–30% in prior years [53,54,55]. Of the IBVs, 97% belonged to subclade V1A.3 as compared with the B/Victoria vaccine reference strain (B/Colorado/06/2017) recommended by the WHO for the 2019–2020 season [54]. The B/Colorado/06/2017-like strain recommended by the WHO contains 2DEL at 162–163, which confers reduced hemagglutinin inhibition cross-reactivity against viruses with 3DEL [56]. However, most of the circulating strains (V1A.3 subclade) contained three amino acid deletions at positions 162–164 of the HA protein sequence [57]. The overall VE for the 2019–2020 season was 46% [57]. Similarly, our study showed that all influenza B strains that were collected in Columbia, Missouri belonged to the Victoria lineage, contained the 3DEL, and were antigenically distinct from the vaccine strain used during this season (Figure 4). As the primary influenza epidemic occurred between February and March of 2020 in this influenza season, the sampling period was somewhat limited, and the inclusion of samples outside of the influenza peak time period may have provided additional insights to the antigenic and genetic evolution of IBVs in this season.

In addition to the antigenic mismatch between the vaccine and circulating influenza B strain due to an unusually early emergence and predominance of IBVs, we demonstrated that 13 hCK cell-adapted mutations were present in the first passage of Missouri strains during the 2019–2020 season. In our samples, no mutations were identified on the antibody binding sites of HA and NA proteins, although a single adaptation was identified at the HA2 protein. However, multiple adaptations in the RdRp were generated during the isolation and propagation of the original wild-type viruses. The effects of mutations in the RdRp have been shown to contribute towards influenza A pathogenicity [58]. However, this process is not well studied in influenza B. The mutations that were identified in the RdRp have not been previously reported, and, as future studies, we plan to explore the antigenic and pathogenic implications of cell-adapted mutations in the RdRp. Further exploration the IBV genomic sequences for the isolates in the GISAID database revealed no or low frequencies for the nonsynonymous mutation site mutations that we observed in the hCK-derived isolates (for example, NP-140 (2 of 11,110 GISAID sequences; 2/11,110), PB2-98 (9/11,121), PB1-350 (1/11,122), and HA-581 (0/11,132)). 

One potential limitation for this study is that we performed analyses for the paired swab only with hCK-derived isolates but not Siat1-, CCL34-, or egg-derived isolates. In addition, the number of paired swab clinical samples (*n* = 31) included in this study is relatively small. Nevertheless, through analyses with those paired samples in the public database, when the entirety of the available Victoria lineage pairs from GISAID were analyzed, cell- and egg-adapted mutations were identified in the majority of protein segments (Table 3; Figure 2). Egg-adapted mutations occurred more frequently than cell-adapted mutations. Multiple reoccurrences appeared at the HA 212 and 214 positions (Figure 2), of which site 212 has been identified as a site of N-glycosylation that undergoes positive selection and frequent mutation (PDB 4FQM) [59]. Cell-adapted mutations at position 212 on HA have also been shown to decrease acid stability in H3N2 viruses in ferret models [60]. Four antigenic sites have been identified in IBVs and are located on the 120-, 150-, and 160-loops and the 190-helix [61]. The 212 and 214 positions reside on the receptor binding domain (RBD) and 190-helix of HA (Figure 2). Additionally, one sequence contained a cell-adapted mutation in the 120-loop (E143K) (Appendix A). Of interest, this observation is different from the mutations in the Missouri swab and hCK samples, which were only present in the internal genes, particularly in the RNA polymerase complex. The HA protein plays a major role in host receptor binding and membrane fusion, and mutations at the RBD and antibody binding sites may affect the antigenic characteristics of and immune responses against IBVs. Culture-adapted mutations during vaccine development at this gene may lead to antigenic variants from the circulating virus.

Analysis of the GISAID database confirms our findings that cell culture-adapted mutations occur in Victoria lineage IBVs, though infrequently. We also found that egg adaptations occurred 5.7 times more often than cell-adapted mutations, and such adapted mutations were reported to be under positive selection due to egg culture [62]. The adapted mutations in HA, particularly those in or close to receptor binding sites, may alter virus receptor binding properties to increase replication in eggs [63]. Similar to our findings, Park et al. showed that H3N2 viruses cultured in eggs had more frequent antigenic mutations in the HA protein than in cells, where they found no antigenic mutations [64]. These culture-adapted mutations have been suggested to compromise vaccine effectiveness for IAVs in humans during multiple influenza seasons [32,33,34,35,36] and remain a concern. Nevertheless, our study showed that using cell culture can help mitigate these challenges.

In summary, our study suggests that cell culture generates less frequent culture-adapted mutations than egg culture for isolating IBVs and that, of the commonly used MDCK cell lines tested, hCK is the most effective platform for IBV isolation. Additionally, culture-adapted mutations may occur in IBVs even after a single passage in hCK cells, and these mutations could generate biases on our understanding of the viral characteristics in the human population. As culture-adapted mutations may affect subsequent virus studies and vaccine development, the knowledge from this study may help optimize strategies for influenza surveillance, vaccine strain selection, and vaccine development.

## Figures and Tables

**Figure 1 viruses-13-01896-f001:**
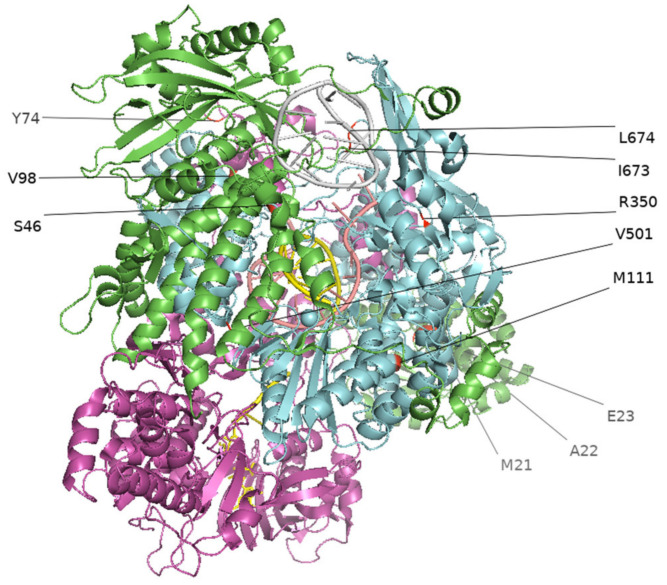
Cell-adapted mutations in the RNA polymerase complex of influenza B virus isolates (2019–2020). Mutations at the RNA polymerase complex (PB1, PB2, PA) are visualized in PyMOL using the 6QCT protein structure from the Protein Data Bank (PDB, https://www.rcsb.org/, accessed on 27 June 2021). Green represents PA, blue represents PB1, purple represents PB2.

**Figure 2 viruses-13-01896-f002:**
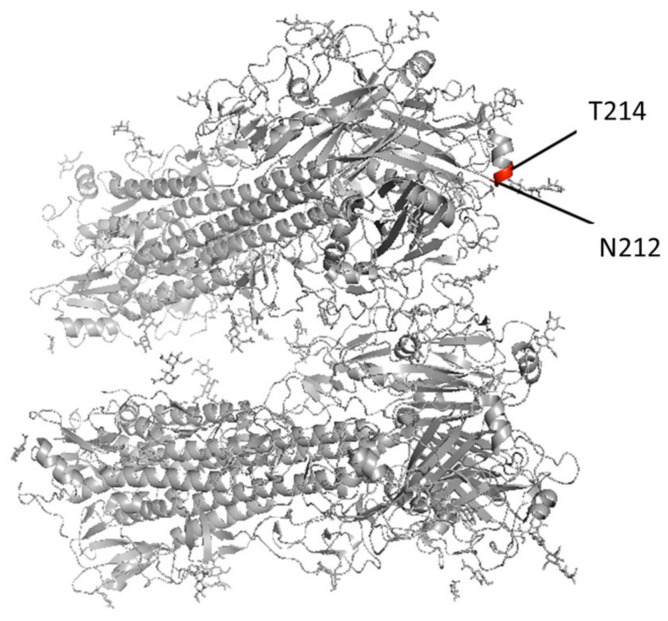
Cell- and egg-adapted mutations at the 212 and 214 positions within the receptor binding domain and antibody binding site on the HA protein visualized in PyMOL using the 4FQM protein structure from the Protein Data Bank (PDB, https://www.rcsb.org/, accessed on 27 June 2021).

**Figure 3 viruses-13-01896-f003:**
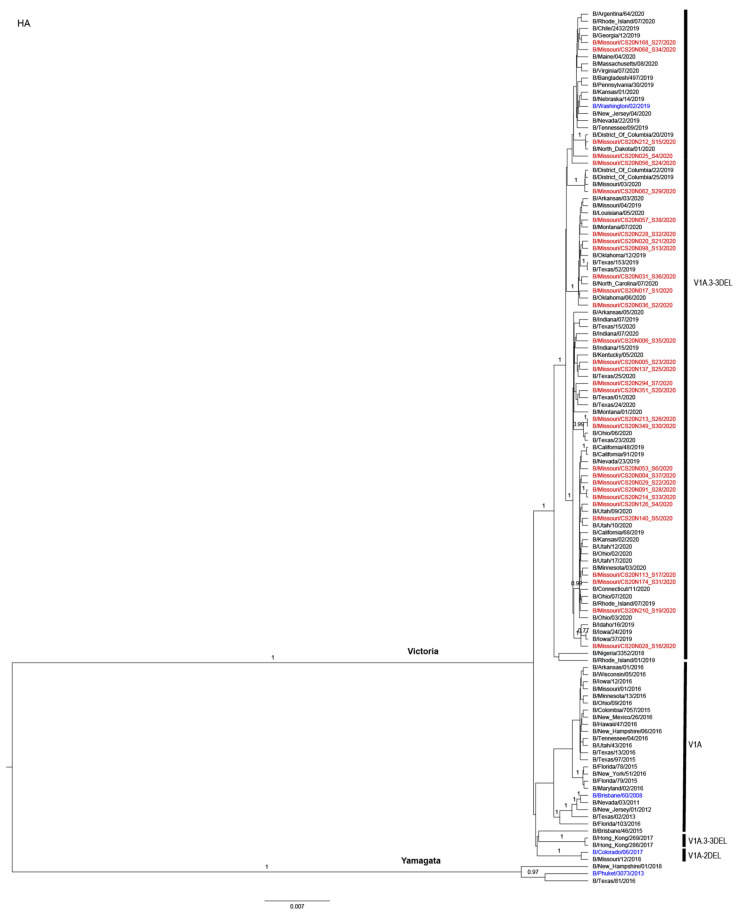
Phylogenetic tree of the HA segment of IBV. Missouri samples from the 2019–2020 season that were collected for this study are labeled in red. Recommended vaccine strains are labeled in blue. The phylogenetic analyses were performed by using BEAST2 with a Hasegawa–Kishino–Yano (HKY) substitution empirical model, strict clock model, and coalescent constant population prior. Branches with a posterior probability ≥0.70 were labeled. The nomenclature of the genetic clades is adapted from that reported by the World Health Organization [51].

**Figure 4 viruses-13-01896-f004:**
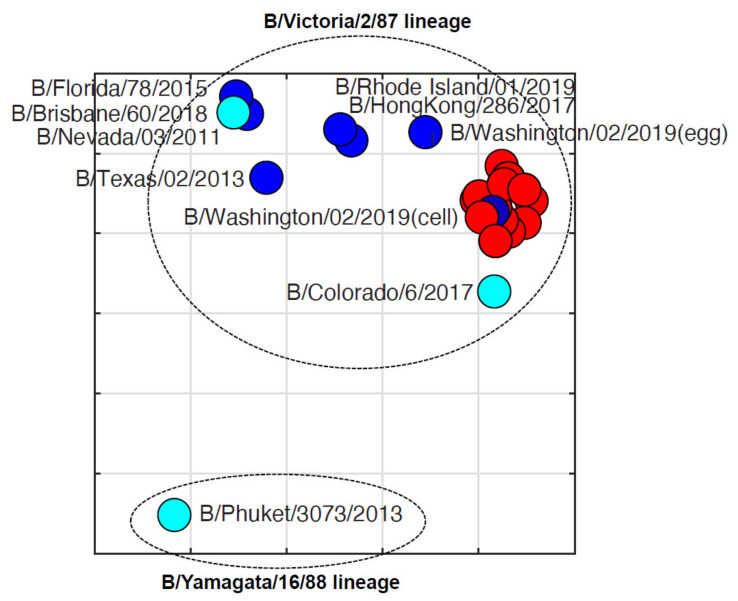
Antigenic cartography of hemagglutination inhibition results of influenza B virus isolates (2019–2020). Each grid represents one unit, a 2-fold change in hemagglutination inhibition titers. The vaccine component strains are labeled in cyan. Strains from the 2019–2020 season in Missouri are shown in red. Additional reference strains are blue. Antigenic cartography was constructed using AntigenMap (https://sysbio.missouri.edu/software/AntigenMap/, accessed on 19 July 2021) [42,43,44].

**Table 1 viruses-13-01896-t001:** Isolation of clinical samples positive to influenza B virus in different MDCK cells.

Type	Samples	CCL34 P1	CCL34 P2	SIAT1 P1	hCK P1
IAV (H1)	108	39 (42.12%)	76 (70.37%)	77 (71.30%)	95 (87.96%)
IBV	38	22 (57.89%)	24 (63.16%)	22 (57.89%)	31 (81.58%)
Total	146	61 (41.78%)	100 (68.49%)	99 (67.80%)	127 (86.98%)

**Table 2 viruses-13-01896-t002:** Cell-adapted mutations in influenza B virus isolates (2019–2020).

Sample	Protein	Position	Clinical Swab (Nucleotide)				Isolate (Nucleotide)					Amino Acid ^e^	
			Swab (Proportion Reads) ^a^	Reads (Total) ^b^	Reads (Position) ^c^	% Reads at Position ^d^	Isolate (Proportion Reads) ^a^	Reads (Total) ^b^	Reads (Position) ^c^	% Reads at Position ^d^	Position	Swab	Isolate
CS20N017	PB2	136	A (0.99)/T (0.01)	1,574,718	26224	1.67	T (0.99)	1,149,216	17,538	1.53	46	**T**	**S**
	PB2	2181	T (0.98)/C (0.02)	1,574,718	7075	0.45	C (0.99)/T (0.01)	1,149,216	54,788	4.77	727	N	N
	PB1	2160	G (1.00)	1,574,718	47919	3.04	A (1.00)	1,149,216	43,524	3.79	720	R	R
CS20N025	HA	1741	C (0.79)/T (0.16)/N (0.05)	1,588,004	45	0.00	T (1.00)	182,972	14,112	7.71	581	**R**	**C**
CS20N036	PB1	2016	T (0.97)/A (0.03)	1,940,088	39	0.00	C (0.50)/T (0.46)/A (0.04)	1,468,024	42,152	2.87	672	S	S
	PB1	2020	T (1.00)	1,940,088	41	0.00	A (0.53)	1,468,024	43,968	3.00	674	**L**	**Q**
CS20N091	NP	419	A (0.96)/N (0.03)/T(0.01)	2,161,170	1615	0.07	C (0.76)/A (0.24)	177,412	4283	2.41	140	**E**	**A**
CS20N137	PB2	293	T (0.96)/N (0.03/A(0.01)	2,078,362	336	0.02	G (0.57)/T (0.43)	216,996	514	0.24	98	**V**	**G**
CS20N140	PB2	2097	T(0.55)/C(0.44)	1,939,620	5992	0.31	C (0.98)/T (0.02)	1,057,874	76,928	7.27	699	L	L
CS20N212	PB1	1048	A (0.96)/N (0.03)/C (0.01)	1,927,990	230	0.01	G (0.62)/A (0.38)	142,724	148	0.10	350	**R**	**G**
	PB1	1502	T (0.94)/G (0.02)/N (0.02)/A(0.01)/C (0.01)	1,927,990	238	0.01	C (0.69)/T (0.31)	142,724	580	0.41	501	**V**	**A**
CS20N214	PB1	255	A (0.77)/G (0.13)/N (0.08)/C (0.01)/T (0.01)	1,779,338	1811	0.10	G (0.56)/A (0.43)/T (0.01)	180,974	59,503	32.88	85	L	L
CS20N294	PB1	335	A (0.99)/C (0.01)	1,495,964	650	0.04	G (0.56)/A (0.43)	1,599,890	12,662	0.79	112	**E**	**G**

^a^ Swab (Proportion Reads) and Isolate (Proportion Reads) refer to the frequency of each nucleotide (A, T, G, or C) compared with the total number of reads (all A, T, G, and C) generated at a specific genomic position. ^b^ Reads (total) refers to the total number of reads generated during whole genome sequencing for each sequence analyzed. ^c^ Reads at Position refers to the number of reads generated by whole genome sequencing at each position of the genome. ^d^ % Reads at Position was calculated by [Reads at Position/Reads (total) × 100]. ^e^ Nonsynonymous mutations are highlighted in bold in the amino acid columns.

**Table 3 viruses-13-01896-t003:** Cell- and egg-adapted amino acid mutations of Victoria lineage IBVs from the paired clinical isolate samples available at the GISAID.

Protein	Strain	Position	Original	Cell	Egg
HA	B/North Carolina/17/2019/EPI1693897	143	E	K	
	B/Florida/78/2015/EPI721064	156	G	G	R
	B/Mozambique/413/2016/EPI854592	179	D	Y	
	B/Pennsylvania/19/2016/EPI807677	212	N	N	K
	B/Michigan/18/2016/EPI807685	212	N	N	T
	B/Indiana/08/2016/EPI807669	212	N	N	D
	B/Florida/94/2016/EPI872715	212	N	N	S
	B/Hawaii/26/2018/EPI1315023	212	N		D
	B/Florida/79/2015/EPI745107	212	N		Y
	B/Utah/18/2018/EPI1249578	212	N	T	
	B/Montana/34/2018/EPI1386557	212	N		D
	B/Iowa/14/2017/EPI1025486	212	T	T	N
	B/Guatemala/35/2018/EPI1249639	212	N	S	
	B/Guatemala/109/2018/EPI1312932	212	N	S	
	B/Florida/97/2017/EPI1138021	212	N		K
	B/Alabama/02/2017/EPI980739	212	N	N	S
	B/Lebanon/16/2020/EPI1840705	212	N	N	S
	B/Pennsylvania/60/2016/EPI892343	214	T		I
	B/Hawaii/55/2017/EPI1089741	214	T		N
	B/Costa Rica/5521/2016/EPI908149	214	T	A	
	B/El Salvador/697/2018/EPI1357148	214	T		A
	B/Louisiana/16/2019/EPI1574236	214	T	T	N
	B/Hong Kong/269/2017/EPI1141721	214	T	T	I
	B/Minnesota/39/2017/EPI1026300	236	I	T	
	B/Ulaabaatar/1868/2017/EPI1089764	433	N	D	
	B/Ulaabaatar/1834/2017/EPI1053102	433	D	N	
	B/Ulaabaatar/1767/2017/EPI1053094	433	N	D	
	B/Wisconsin/22/2018/EPI1312153	528	D	G	
NA	B/North Carolina/17/2019/EPI1693896	19	L	I	
	B/North Carolina/16/2019/EPI1664425	19	I	L	
	B/Wisconsin/45/2016/EPI834520	154	L	K	
	B/North Carolina/17/2019/EPI1693896	284	S	G	
	B/North Carolina/16/2019/EPI1664425	284	G	S	
	B/Ulaabaatar/1868/2017/EPI1089763	310	T	S	
	B/Nevada/23/2019/EPI1688225	321	D	N	
	B/Colombia/9962/2016/EPI908031	391	D	E	
	B/Bangladesh/8018/2016/EPI880659	395	A	D	
	B/North Carolina/17/2019/EPI1693896	400	V	I	
	B/Arizona/83/2017/EPI1147826	401	M	V	
	B/California/18/2017/EPI1009716	435	G	R	
	B/Guyane/310/2020/EPI1807176	450	M	M	V
NB	B/North Carolina/17/2019/EPI1693896	21	I	N	
	B/North Carolina/16/2019/EPI1664425	21	N	I	
NEP	B/Oregon/11/2020/EPI1840666	30	S	S	L
	B/Iowa/14/2017/EPI1011596	79	D	D	N
NP	B/Bangladesh/5006/2017/EPI1094384	33	R	K	
	B/Ulaabaatar/1868/2017/EPI1089757	239	A	T	
NS1	B/North Carolina/17/2019/EPI1693891	62	T	A	
	B/North Carolina/16/2019/EPI1664420	62	A	T	
	B/Ulaabaatar/1868/2017/EPI1089758	115	T	M	
	B/Ulaabaatar/1834/2017/EPI1053096	115	M	T	
	B/Ulaabaatar/1767/2017/EPI1089797	115	T	M	
PA	B/Bangladesh/8018/2016/EPI880656	57	E	K	
	B/New Jersey/09/2017/EPI1011540	393	I	I	T
PB1	B/North Carolina/17/2019/EPI1693895	378	L	S	
	B/North Carolina/16/2019/EPI1664424	378	S	L	
PB2	B/Florida/97/2017/EPI1138018	263	V		I
	B/Macedonia/421/2020/EPI1806732	379	S	S	C
	B/Bangladesh/5006/2017/EPI1094388	391	A	T	

Blank cells under cell and egg mutation columns indicate that no sequence was available for analysis.

**Table 4 viruses-13-01896-t004:** Hemagglutination inhibition results of influenza B virus isolates (2019–2020) using ferret antisera.

Virus ^a^	Clade ^b^	Ferret Reference Sera ^c^						
		B/Nevada/ 03/2011	B/Texas/ 02/2013	B/Brisbane/ 60/2008	B/Colorado/ 6/2017	B/Washington/ 02/2019 (Egg)	B/Washington/ 02/2019 (Cell)	B/Wisconsin/1/2010
B/Nevada/03/2011	V1A	**1280**	320	640	20	160	80	10
B/New Jersey/01/2012	V1A	1280	640	320	40	40	40	<10
B/Texas/02/2013	V1A	320	**320**	160	40	40	40	<10
B/Florida/78/2015	V1A	1280	1280	640	80	80	80	20
B/Brisbane/60/2018	V1A	640	320	1280	40	160	40	10
B/Colorado/06/2017	V1A-2DEL	40	20	0	**320**	40	160	<10
B/Hong Kong/286/2017	V1A.3-3DEL	320	160	160	80	320	160	10
B/Washington/02/2019 (egg)	V1A.3-3DEL	320	80	80	80	**640**	640	<10
B/Washington/02/2019 (cell)	V1A.3-3DEL	80	40	10	80	320	**640**	<10
B/Rhode Island/01/2019	V1A.3-3DEL	320	160	320	40	320	320	10
B/Phuket/3073/2013	Y3	10	<10	<10	<10	<10	<10	**1280**
CS20N004	V1A.3-3DEL	80	40	10	160	320	640	<10
CS20N005		80	40	10	160	320	640	<10
CS20N006		80	80	20	160	640	1280	<10
CS20N017		80	40	10	160	320	640	<10
CS20N020		80	40	10	160	640	1280	<10
CS20N025		80	40	20	160	640	1280	10
CS20N028		80	40	10	160	320	640	<10
CS20N029		80	40	20	160	640	1280	<10
CS20N031		80	40	10	160	320	640	<10
CS20N036		80	40	20	160	640	1280	10
CS20N053		80	40	10	160	640	1280	<10
CS20N056		80	40	20	160	640	1280	<10
CS20N057		40	20	10	80	320	640	<10
CS20N062		40	40	10	160	640	1280	<10
CS20N068		40	20	10	80	640	1280	<10
CS20N091		80	40	10	160	320	640	<10
CS20N098		40	20	10	80	320	640	<10
CS20N113		40	20	10	160	320	640	<10
CS20N126		40	20	10	80	320	640	<10
CS20N137		80	20	10	160	320	640	<10
CS20N140		80	20	10	160	320	640	<10
CS20N168		80	40	20	160	640	1280	<10
CS20N174		80	40	10	160	320	640	<10
CS20N210		80	40	20	160	320	640	10
CS20N212		80	40	20	160	640	1280	<10
CS20N213		80	40	20	160	320	640	<10
CS20N214		80	40	20	160	320	640	<10
CS20N228		80	40	20	160	640	1280	10
CS20N274		80	40	10	160	320	640	10
CS20N294		80	40	10	80	320	640	<10
CS20N349		40	20	10	80	320	640	<10
CS20N351		80	40	10	160	640	1280	<10

^a^ The viruses underlined are the vaccine strains used in the 2019–2020 influenza season. Passage 2 viruses for all Missouri isolates were used in the serological assays. ^b^ The 3DEL noted in clades occurs at positions 162–164 on the HA protein. ^c^ The homologous hemagglutination inhibition titers were marked in bold.

## Data Availability

The data presented in this study are openly available in the GenBank (accession numbers: MZ951177–MZ951671).

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
