# Peer review of "Cell-Adapted Mutations and Antigenic Diversity of Influenza B Viruses in Missouri, 2019–2020 Season"

_viruses, 2021, doi:10.3390/v13101896_

Round 1
Reviewer 1 Report
This study aims to capture the cell-adaptive mutations of influenza B viruses. To do this the authors screen and characterise IBV in Missouri, US during February-March 2020, assess their isolation in cells, investigate cell-adaptive mutations and perform genetic and antigenic characterisation of the viruses. While the authors provide a detailed characterisation of the viruses, the central question (of understanding cell-adaptive mutations) is affected due to the limited diversity of viruses used from only one season. In addition, the focus of the paper seems wavering (see below), and not one aspect is covered in detail. There are also several incorrect statements in the ms. In summary, I have several major concerns and a thorough revision is required prior to consideration.
1. There is a disconnect between the title and abstract and the rest of the article. From the title and abstract, it appears that the study aims to capture the cell-adaptive mutations of IBV, however on reading the rest it appears that only a very small portion of the study address this passingly, whilst the main focus of the study lies elsewhere. This is misleading and the ms might require a re-write (also see below).
2. Further, the introduction contains several sentences that are simply wrong, and some of the concepts are incorrectly stated or not clear. Some examples:
- Why would the authors expect B/Yamagata and B/Victoria to cause pandemics as stated in the first paragraph of the introduction “neither have caused a pandemic to date”. They are already circulating in humans?
- In another sentence the authors state “The lesser pandemic potential of IBVs is due to their slower evolution compared with IAVs [11-13].” This is completely wrong. IBV cannot cause pandemics because they don’t have an animal reservoir.
- Authors state IBV cause a ‘disproportionally high burden of mortality, particularly in children.’ and cite three papers. However, while one of the papers does state *disproportionate* for children, but the data suggests it is only higher in China and certainly NOT disproportionate. Also as this is not the focus of the paper, the authors should avoid perpetuating such tall claims passingly.
- The authors state “Antigenic drift is responsible for the annual recurrence of influenza epidemics [20]. “ While antigenic drift enables the continued circulation of individual flu lineages, certainly there are many factors affecting seasonal/annual epidemics. Also this statement discounts the complexity of flu in tropical, and sub-tropical regions, and environmental factors.
- It is simplistic to call antigenic analysis a ‘serological test’ in the introduction.
3. Unfortunately, as the study design uses only viruses collected during one or two months in 2020, the authors were only able to test a small diversity of B/Victoria viruses for cell-adaptations, making the main goals of the study incomplete.
4. Title and abstract say 2019-2020, but only February - March 2020 is stated in the results.
5. The phylogenies used for reassortment analysis should be improved. The visual (Figure 4) suggests reassortment of monophyletic clades, but on closer inspection, this appears to be misleading. Only two strains from Missouri appear to be reassortants, which no evidence of further transmission, suggesting they could indeed be co-infections. Also, it is better to construct phylogenies of B/Victoria viruses separately (without B/Yamagata) to better infer reassortment patterns.
6. It is not clear what the antigenic analysis adds to this ms.
Reviewer 2 Report
In the study by Tang et al the frequency of mutations in Influenza B virus proteins of Missouri 2019-2020 isolates were evaluated after in vitro culture in various cells lines as well as eggs to advance our understanding as to how to best minimize vaccine production mismatches.
Comments on the manuscript in its current form are as follows:
- The abstract is missing a statement as to why such research is important.
- Only one cell line, the humanized MDCK, is extensively evaluated for the frequency of mutations detected. How does the frequency and nature of the mutations identified compare among the other cell lines?
- All mutations detected within the isolates were detected in the progeny of the first passage of virus. There is no guarantee that the mutated viruses are indeed infectious and viable and would end up being used in vaccine production. What measures were taken to remove defective interfering particles from the study?
- In the methods, it should be stated how many total reads were done and in Table 2, inclusion of the total mutant reads detected as a percentage of the total reads would be helpful.
- It is stated within the results in section 3.2 that only samples with complete sequences at each influenza segment were included within the analysis. Detailed information as to the number of complete versus incomplete sequences should be included to give one an idea of the depth and breadth of the sequencing of the isolates.
- The appendix tables contain essential information and should be included within the main body of the manuscript whereas the phylogenetic trees more fitting in the supplemental information.
- The majority of the mutation reads detected in the passaged Missouri isolates fall within the internal viral proteins that are well known to be highly conserved and only few reads (45) with mutations detected in the coat proteins known to be the most variable in influenza viruses. The authors should include a brief discussion regarding this, any differences between this data and the GISAID strains in the Appendix tables, and the impact of this balance of mutations on eventual vaccine production.
Round 2
Reviewer 1 Report
Cell-adaption during vaccine production is a concern that has been particularly shown for A(H3N2) viruses, and less known for IBV. By comparing the growth of 38 IBV isolates (collected by authors in Missouri during 2019-2020 season) in three MDCK cell culture types, the authors identify highest virus recovery in humanised MDCK. This message is coming through more clearly in the revised version. However, the subsequent sections on cell-adaptation mutations can be improved significantly, and ultimately I could not determine what the main message for this study is. Are IBV cell-adaption mutations are a concern or not?
1. Critically, results for the core part of the study (section 3.2 on cell adaptation mutations) is very brief in the manuscript. “From these 31 paired samples, we identified eight nonsynonymous mutations among these paired samples that have not been previously published (HA-R581C; PB2-T46S, PB2-V98G; PB1-E112G, PB1-R350G, PB1-V501A, PB1-L674Q; and NP-E140A), six of which were from RNA polymerase complex (Figure 1), and six synonymous mutations (Table 2). Polymorphisms were also identified in the PB2-T293G, PB2-T2097C, PB2-T2181C, PB1-A255G, PB1-A1048G, PB1-T1502C, and NP-A419C (Table 2).”
Below I have listed a few suggestions for authors' consideration.
(a) It would be better if this paragraph included some inferences and not just a list of amino acids. For instance, add something to describe the details in Table 2.
Suggestion: “Thirteen non-synonymous cell-adaptation mutations were detected in total among nine viruses (range, one to three mutations per virus), and all of the mutations were unique to each isolate. ..”.
(b) Better to mention here that *no* antigenically relevant mutations were identified in the study? Even the HA mutation detected in a single isolates looks like in the HA2 region (nt 1741)? Add some discussion?
(c) Can you say something about all the NGS reads data presented in Table 2?
(d) Analysis of the GISAID IBV data can also be improved. The frequencies of these mutations are so low, that I am not sure what is key take-away message were. Based on this analysis can you say the frequency of mutation in IBV is low (e.g. in comparison to H3N2)? How does the frequency of mutations differ between IBV strains and IAV strains reported in previous sudies. Further sentences in Results/Discussion is needed.
(e) What is the differences in virus passage between this study and those presented in GISAID?
2. Revise lines 23-24 (Abstract) to clarify that the “three-amino acid deletion” are present in all IBV from this lineage, and not just restricted to Missouri, nor to cell cultures mentioned in the previous sentence (to avoid confusion).
Author Response
Response to Reviewer:
Cell-adaption during vaccine production is a concern that has been particularly shown for A(H3N2) viruses, and less known for IBV. By comparing the growth of 38 IBV isolates (collected by authors in Missouri during 2019-2020 season) in three MDCK cell culture types, the authors identify highest virus recovery in humanised MDCK. This message is coming through more clearly in the revised version. However, the subsequent sections on cell-adaptation mutations can be improved significantly, and ultimately I could not determine what the main message for this study is. Are IBV cell-adaption mutations are a concern or not?
Response: Thank you again for these constructive comments. Through this study, we showed that culture-adapted mutations occur during virus isolation but less frequently when using hCK cells. Culture-adapted mutations can occur in antibody binding sites of HA, as seen in public databases, whereas culture-adapted mutations were only observed in HA2 and internal proteins in this study using Missouri viruses when hCK cells were used in virus isolation. Thus, the IBV culture adapted mutations will continue to be a concern, and selection of an optimal platform for virus isolation is important. In addition, antigenicity-associated mutations may generate biases in vaccine strain selection, and any of these culture-adapted mutations, including those we observed in hCK cells, may generate biases for understanding the viral characteristics in humans.
- Critically, results for the core part of the study (section 3.2 on cell adaptation mutations) is very brief in the manuscript. “From these 31 paired samples, we identified eight nonsynonymous mutations among these paired samples that have not been previously published (HA-R581C; PB2-T46S, PB2-V98G; PB1-E112G, PB1-R350G, PB1-V501A, PB1-L674Q; and NP-E140A), six of which were from RNA polymerase complex (Figure 1), and six synonymous mutations (Table 2). Polymorphisms were also identified in the PB2-T293G, PB2-T2097C, PB2-T2181C, PB1-A255G, PB1-A1048G, PB1-T1502C, and NP-A419C (Table 2).”
Below I have listed a few suggestions for authors' consideration.
(a) It would be better if this paragraph included some inferences and not just a list of amino acids. For instance, add something to describe the details in Table 2.
Suggestion: “Thirteen non-synonymous cell-adaptation mutations were detected in total among nine viruses (range, one to three mutations per virus), and all of the mutations were unique to each isolate...”.
Response: We thank the reviewer for their constructive comments. This statement has been updated in section 3.2.
(b) Better to mention here that *no* antigenically relevant mutations were identified in the study? Even the HA mutation detected in a single isolate looks like in the HA2 region (nt 1741)? Add some discussion?
Response: We have expanded the discussion on antigenically relevant mutations identified in the study in the discussion section of the manuscript. With hCK cells as the virus isolation platform, no mutations were identified in HA1; however, when with eggs or other cell platforms, a number of mutations were observed in HA1, including those in antibody binding sites, which are indeed antigenically relevant. We also expanded our discussion on the biological significance for those mutations in the virus proteins other than HA, such as those we identified in the RNA polymerase complex, which could be associated with influenza virus-host adaption and pathogenicity. On the other hand, we do understand the sample size in this study is small, and we added this as a limitation for this study.
(c) Can you say something about all the NGS reads data presented in Table 2?
Response: Discussion of the NGS reads data have been added to the legend in Table 2.
(d) Analysis of the GISAID IBV data can also be improved. The frequencies of these mutations are so low, that I am not sure what is key take-away message were. Based on this analysis can you say the frequency of mutation in IBV is low (e.g. in comparison to H3N2)? How does the frequency of mutations differ between IBV strains and IAV strains reported in previous studies. Further sentences in Results/Discussion is needed.
Response: We thank the reviewer for this valuable suggestion. We have included additional discussion of the GISAID analysis and comparisons between IBV and IAV findings.
(e) What is the differences in virus passage between this study and those presented in GISAID?
Response: We agree with the reviewer that it would be important to consider differences in mutations with each subsequent passage, and a comparison between P1 of both our samples and samples in the GISAID database would be valuable. Unfortunately, the GISAID database does not have detailed passage number, and thus, we are unable to stratify by passage number in our analysis using the GISAID sequences. Thus, the analyses only focused on whether or not culture-adapted mutations occur when using those platforms. We have clarified this in section 3.2.
- Revise lines 23-24 (Abstract) to clarify that the “three-amino acid deletion” are present in all IBV from this lineage, and not just restricted to Missouri, nor to cell cultures mentioned in the previous sentence (to avoid confusion).
Response: This clarification has been updated in the abstract.

Round 3
Reviewer 1 Report
The authors addressed all of my comments.